# Evaluating alignment between humans and neural network representations in image-based learning tasks

**Can Demircan**[1,2,*]    **Tankred Saanum**[2]    **Leonardo Pettini**[3,4]    **Marcel Binz**[1,2]
**Blazej M Baczkowski**[4,5]    **Christian F Doeller**[4,6,7,8]    **Mona M Garvert**[4,9,10]    **Eric Schulz**[1,2]

[1]Institute for Human-Centered AI, Helmholtz Computational Health Center, Munich, Germany
[2]Max Planck Institute for Biological Cybernetics - Tübingen, Germany
[3]Max Planck School of Cognition - Leipzig, Germany
[4]Max Planck Institute for Human Cognitive & Brain Sciences - Leipzig, Germany
[5]University of Tübingen - Tübingen, Germany
[6]Kavli Institute for Systems Neuroscience - Trondheim, Norway
[7]Leipzig University - Leipzig, Germany
[8]Technical University Dresden - Dresden, Germany
[9]Julius-Maximilians-Universität Würzburg - Würzburg, Germany
[10]Max Planck Institute for Human Development - Berlin, Germany
*{can.demircan@helmholtz-munich.de}

## Abstract

Humans represent scenes and objects in rich feature spaces, carrying information that allows us to generalise about category memberships and abstract functions with few examples. What determines whether a neural network model generalises like a human? We tested how well the representations of $86$ pretrained neural network models mapped to human learning trajectories across two tasks where humans had to learn continuous relationships and categories of natural images. In these tasks, both human participants and neural networks successfully identified the relevant stimulus features within a few trials, demonstrating effective generalisation. We found that while training dataset size was a core determinant of alignment with human choices, contrastive training with multi-modal data (text and imagery) was a common feature of currently publicly available models that predicted human generalisation. Intrinsic dimensionality of representations had different effects on alignment for different model types. Lastly, we tested three sets of human-aligned representations and found no consistent improvements in predictive accuracy compared to the baselines. In conclusion, pretrained neural networks can serve to extract representations for cognitive models, as they appear to capture some fundamental aspects of cognition that are transferable across tasks. Both our paradigms and modelling approach offer a novel way to quantify alignment between neural networks and humans and extend cognitive science into more naturalistic domains.

## 1 Introduction

Research on representational alignment between neural networks and humans has gained significant attention in recent years [1, 2]. Comparisons across the systems have provided important insights into neural network representations [3, 4], human cognition and the brain [5, 6, 7, 8, 9], and the development of more robust machine learning systems [10, 11, 12]. In the sensory domain, the comparisons have been predominantly made through two families of behavioural tasks. One common approach is to compare object recognition performance across humans and neural networks [13]. This

38th Conference on Neural Information Processing Systems (NeurIPS 2024).

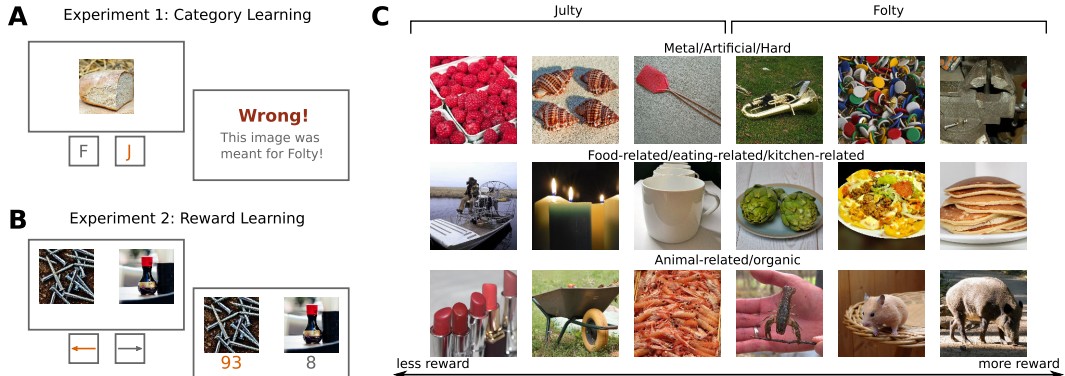

Figure 1: Task descriptions. (*A*) An example trial from the category learning task, where an incorrect decision is made. (*B*) An example trial from the reward learning task where the best option is chosen and highlighted in orange. (*C*) Example images from the THINGS database [30]. The database has a low dimensional semantically interpretable embedding [27], which is derived from human similarity judgements. The example images are placed in the most three prominent dimensions of this embedding. In both tasks, participants were randomly assigned to one of these three dimensions. The associated category membership and rewards for the two tasks are displayed.

is a fruitful approach for understanding if the two systems use the same features for object recognition [14, 15, 16], are susceptible to similar distortions [17, 18, 19, 20], and struggle with similar images [21]. Another common approach is to use similarity judgement tasks, which may entail reporting pairwise similarity scores [22, 23, 24], arranging stimuli in a 2D space based on their similarity [25, 26], or choosing the odd-one-out in triplets of stimuli [27, 28]. Using these tasks, previous work has identified the factors that contribute to neural networks representing stimuli similarly to humans, both in low-level perception [29] and semantic judgements [3].

However, similarity judgements do not begin to capture the complexity of tasks humans use their representations for. Humans rely on rich representations for making judgements and acting in the world. For example, an apple has a multitude of features, such as colour, taste, shape, and brand. Depending on the context, people can use these features and make predictions about the apple's taste, the environmental impacts of growing it, or the significance of it in different mythological and religious settings. What determines whether a neural network model represents an object like an apple with the same richness and flexibility?

In this work, we investigated people's ability to learn functional relationships on naturalistic images in a few-shot setting, and what neural network models best predict human choices. We adapted two commonly used learning paradigms from the cognitive psychology literature: category learning (Fig. 1A) and reward learning (Fig. 1B). However, instead of using repeating artificial stimuli, we presented human participants with unique naturalistic images sampled from the THINGS database [30] in each trial, requiring them to continuously generalise. To understand whether neural networks contain sufficiently rich representations that allow for such generalisation, we tested 86 different neural networks [27, 31, 32, 33, 34, 35, 11, 36, 37, 38, 39, 40, 41, 42, 43, 44, 45, 46, 47, 48, 49, 50, 51, 10, 12, 52, 53, 54, 55]. These networks varied in their loss function, training diet, and the modality of training data. In summary, we found that:

- While almost all pretrained models generalised above chance level and predicted human behaviour in both naturalistic learning tasks, contrastive language image pretraining (CLIP) [45] consistently yielded the best predictions of human behaviour. We furthermore showed that this could not be fully attributed to the training diet alone.

- Multiple factors were important for human alignment, including task performance, model size, training diet, separation of different classes in representations, and the similarity of the representations to the generative embedding of the task.

- Of the tested human-aligned neural networks, no method consistently improved human alignment in our tasks compared to non-aligned baselines. However, two of the methods (Harmonization [11] and gLocal [10]) yielded improvements in task accuracy on average.

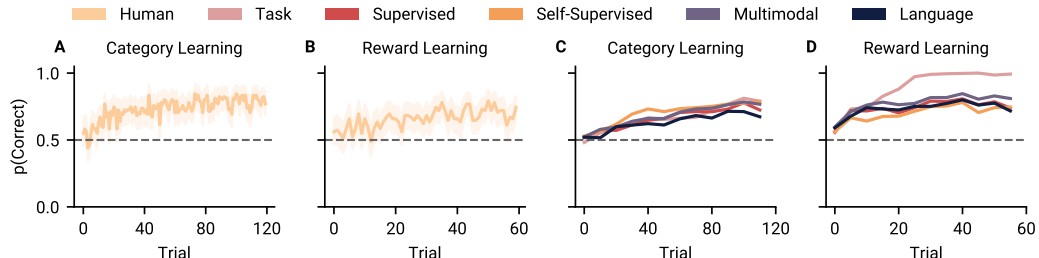

Figure 2: Learning trajectories of human participants and neural networks. Neural networks can perform as well as humans. (*A & B*) Accuracy of human participants across trials for the category and the reward learning tasks respectively. Shaded lines indicate $95\%$ confidence intervals. (*C & D*) Example learning curves for the neural network representations in the category and the reward learning tasks respectively. The best-performing models from each model type are shown.

## 2   Experiments

We design our experiments around naturalistic images from the THINGS database [30, 27]. Each image in the database depicts a collection of entities (animals or objects) and comes with an embedding with 49 human interpretable features, which was built by Hebart et al. [27] to predict human similarity judgements of these objects. Each feature reflects a semantically meaningful property such as whether an image contains metallic objects, food, animals etc. In our experiments, humans learned functions defined over these individual embedding dimensions. We chose category learning and reward learning experiments, as they are well-established paradigms to test function learning and generalisation in human participants. However, unlike traditional paradigms, we used naturalistic images and no images were repeated, requiring generalisation.

**Category learning:** Human participants ($n = 91$) completed 120 trials of an online category learning task, where they were presented with a novel image in each trial. They were asked to deliver these images to one of two dinosaurs, *Julty* or *Folty*, using key presses. Participants were told that the two dinosaurs had completely non-overlapping preferences for what gifts they enjoyed. After each trial, we gave participants feedback on whether their choice of delivery was correct. An example trial from the task is shown in Fig. 1*A*. Participants were assigned to one of three conditions, where in each condition the category boundary was defined over a different THINGS embedding dimension. The three chosen dimensions map to how metallic, food-related, and animal-related the shown image is (Fig. 1*C*). For instance, in one condition non-metallic images should be classified to Folty, and metallic images to Jolty. For each participant, 120 unique stimuli from the THINGS database were sampled. A median split over the assigned feature of the sampled stimuli determined the category boundary.

**Reward learning:** Human participants ($n = 82$) completed 60 trials of a reward learning paradigm [56], in which they were asked to maximise their accumulated reward throughout the task. In each trial, participants were presented with two images and were asked to select one using key presses. After making a choice, the associated reward with each option was shown. An example trial from the task is shown in Fig. 1*B*. Participants were assigned to one of the three conditions, as was done in the category learning task. Stimuli were sampled in the same way as the category learning task. For each participant, the values of the task-relevant feature were re-scaled linearly between 0 and 100. Additional details about the experimental paradigms are described in Appendix A.

## 3   Behavioural analyses

**Humans learn to generalise quickly.** The learning curves of the participants are shown in Fig. 2A and 2B. To measure whether and how fast people learned in the two experiments, we analysed their choice data using mixed-effects logistic regression models. In the category learning task, we predicted whether a participant made the correct choice using an intercept and the trial number. We found that participants performed this task above chance level, as indicated by a significant

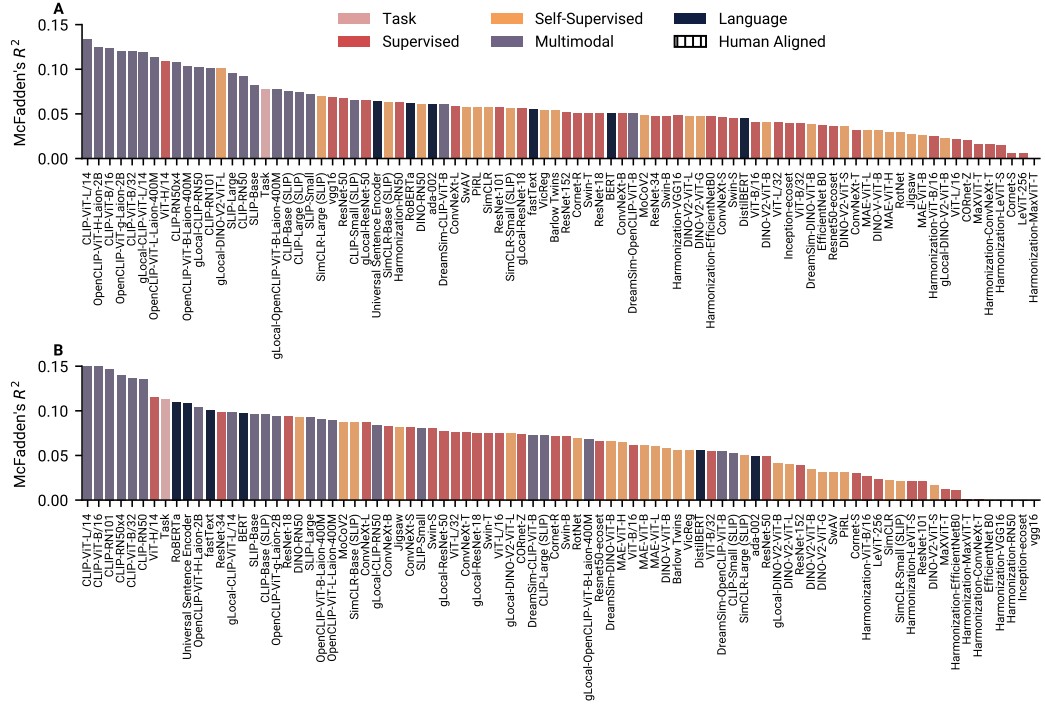

Figure 3: Model fits to human choice data. In both category learning (*A*) and reward learning tasks (*B*), several CLIP models predict human choices the best, even better than the generative features of the tasks. How well the models fitted human choice was more heterogeneously distributed for supervised, self-supervised, and language models. Plotted are the cross-validated McFadden's $R^2$ of each representation for the category learning and the reward learning tasks respectively. Higher values indicate better fits to human behaviour. 0 marks the alignment of a random model.

intercept ($\hat{\beta} = 1.14 \pm 0.09$, $z = 13.18$, $P < .001$), and that their performance improved over trials ($\hat{\beta} = 0.32 \pm 0.05$, $z = 6.89$, $P < .001$), indicating a learning effect. This suggests that people can very efficiently extract the relevant feature dimension in high-dimensional naturalistic environments despite seeing each stimulus only once. For the reward learning task, we predicted whether a participant chose the image on the right using the reward difference between the two images, the trial number, and the interaction of the two predictors. We found that the reward difference ($\hat{\beta} = 0.89 \pm 0.07$, $z = 12.56$, $P < .001$), and the interaction of this difference with the trial number ($\hat{\beta} = 0.34 \pm 0.04$, $z = 9.30$, $P < .001$) predicted choice, again indicating a learning effect. We further characterise how quickly humans learn the task in Fig. 10 in Appendix C and provide the full specification of the mixed-effects models in Appendix A.

## 4   Model-based analyses

To understand what kind of representations are needed to predict human choices, we tested representations extracted from several pretrained neural networks on our tasks.

**Most representations predict human choice above chance level. CLIP makes the best predictions.**
The representations were extracted from the penultimate layer if the models had a classification layer, and from the final layer otherwise. For the transformer models, the `[CLS]` token representations were extracted. To extract representations from language models, we provided them with the prompt `A photo of` $X$ where $X$ was the category label of the task image. fastText was only provided with the category label instead.

We trained linear models to predict either reward or category membership from each neural network model's extracted representations. The models were provided with image-target pairs until trial

$t - 1$ as training data and made predictions for the image on trial $t$. For the category learning task, we used an $\ell_2$ regularised logistic regression model, and for the reward learning task, we used a Bayesian linear regression model with spherical Gaussian priors. We used the estimates from the linear models to predict participant choice using mixed-effects logistic regression in leave-one-trial-out cross-validation. For the category learning task, we regressed the probability estimates of the logistic regression models onto participant choice. For the reward learning task, we regressed the reward estimate differences between the left and the right options onto choice. Example learning curves for the two tasks are shown in Fig. 2C and Fig. 2D. Finally, we measure alignment to human choices using McFadden's $R^2$ [57], which is computed as follows:

$$\text{McFadden's } R^2 = 1 - \frac{\mathcal{L}_{\text{Model}}}{\mathcal{L}_{\text{Random}}} \tag{1}$$

where $\mathcal{L}_{\text{Model}}$ is the negative log likelihood of a given model and $\mathcal{L}_{\text{Random}}$ is the negative log likelihood of a random model.

We observed most of the representations we tested can do our task and predict human behaviour above chance level across the two tasks (as visualized in Fig. 3A and Fig. 3B). CLIP models were the top 7 (6) models for the category (reward) learning task in predicting human choices. In total, 16 (7) of the 86 candidate representations predicted participant behaviour better than the ground truth representations that were used to generate the task. Of these 16 (7) representations, 14 (6) were CLIP models. One was a large vision transformer, trained in a supervised manner on ImageNet [58]. A human-aligned variant of DINO-v2 provided a better fit than the generative task representations in the category learning task. The rest of the supervised and self-supervised vision models, as well as the language models, had a heterogeneous distribution in how well they predicted human behaviour. To provide better intuition for how human participants and CLIP were similar, we display example trials where both CLIP and humans make the same incorrect decisions in Fig. 11 in Appendix C.

**Which factors contribute to alignment?**

Why are CLIP models substantially better aligned with humans in our task? We conducted a series of analyses to better understand which model properties contribute to alignment. We pooled the data across the two tasks and excluded the language models from all analyses except those shown in Fig. 4A and Fig. 4E, as comparing other properties across vision and language models (e.g. model size) is not meaningful. We first tested if larger models predicted human choice better. While it is common for more expressive models to perform better at downstream computer vision tasks [59, 60, 61], previous work has shown that this is not a robust predictor of human alignment [3, 9]. In our tasks, we found that larger models predicted human choices better ($\rho = 0.48$, $p < .001$, Fig. 4B), which contradicts previous findings. Next, we considered the number of images seen during training, which is predictive of higher accuracy in image recognition [62] and human alignment. In our tasks, we found that models trained on more images were more predictive of human choices as well ($\rho = 0.52$, $p < .001$, Fig. 4C).

Then, we analysed which, if any, properties of the models' representations were predictive of their alignment with human choices. First, we considered how well the THINGS classes were separated in the representations of each model. Following Kornblith et al. [63], the class-separation was computed as follows:

$$R^2 = 1 - \bar{d}_{\text{within}} / \bar{d}_{\text{total}} \tag{2}$$

$$\bar{d}_{\text{within}} = \sum_{k=1}^{K} \sum_{m=1}^{N_k} \sum_{n=1}^{N_k} \frac{1 - \cos(\mathbf{x}_{k,m}, \mathbf{x}_{k,n})}{K N_k^2} \quad \bar{d}_{\text{total}} = \sum_{j=1}^{K} \sum_{k=1}^{K} \sum_{m=1}^{N_j} \sum_{n=1}^{N_k} \frac{1 - \cos(\mathbf{x}_{j,m}, \mathbf{x}_{k,n})}{K^2 N_j N_k} \tag{3}$$

where $\mathbf{x}_{k,m}$ is the representation of image $m$ in object class $k$. $K$ is the total number of classes, and $N_k$ is the total number of images in class $k$. $\cos(\cdot, \cdot)$ denotes cosine similarity between representations. The $R^2$ measure is between 0 and 1, where higher scores indicate a low within-class distance to across-class distance ratio, i.e. high class separation. Previous work has shown a positive link between class separation and image classification [63], as well as recall [64]. Similar to these findings, we found that models that had higher class separation were more predictive of human choices ($\rho = 0.29$, $p = .01$, Fig. 4D).

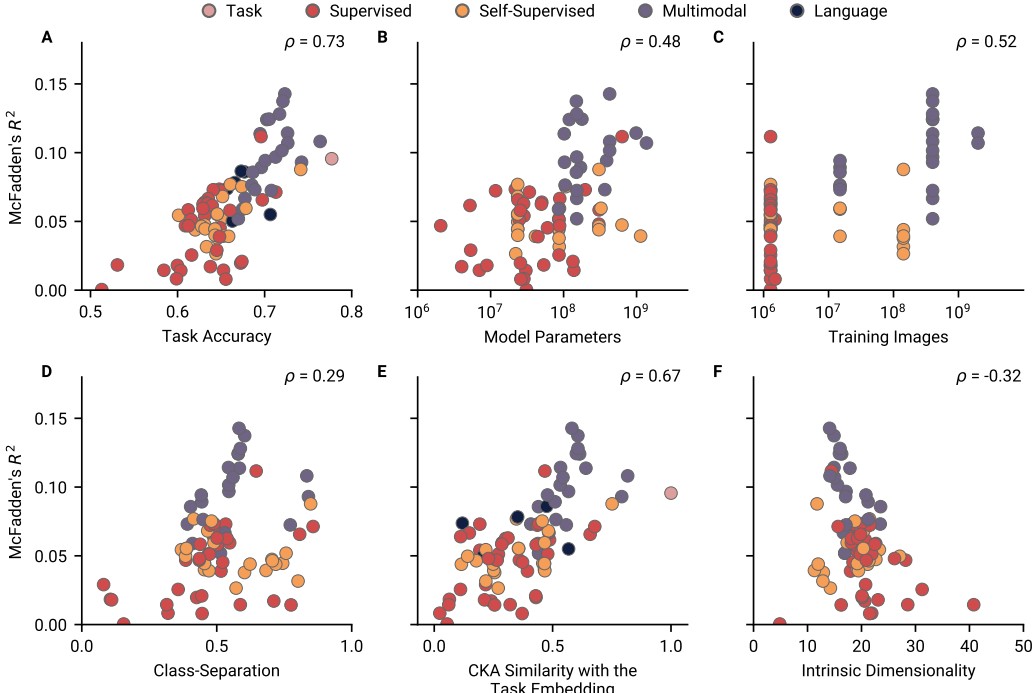

Figure 4: Several factors contribute to alignment. Models trained on more data and with more trainable parameters predict human choices with higher accuracy. Turning to representations, those that better separate image classes and are more similar to the generative task features exhibit stronger alignment with human choices.

We then considered whether the similarity of the representations with the generative task features was predictive of how well different representations predicted human choices. For this, we used linear Centered Kernel Alignment (CKA) [65], which computes the similarity between the generative task representations $\mathbf{T}$ and neural network representations $\mathbf{X}$ as follows:

$$\mathrm{CKA}(\mathbf{T}, \mathbf{X}) = \frac{||\mathbf{X}^T\mathbf{T}||_F^2}{||\mathbf{T}^T\mathbf{T}||_F||\mathbf{X}^T\mathbf{X}||_F} \tag{4}$$

where $|| \cdot ||_F$ denotes the Frobenius norm. We found that representations that were more similar to the generative task embedding predicted human choices better ($\rho = 0.67$, $p < .001$, Fig. 4E).

Lastly, we tested whether the intrinsic dimensionality of representations was related to alignment. Lower intrinsic dimensionality of neural networks in late layers is positively linked to better classification performance [66]. The degree to which a network *compresses* its inputs is also directly linked to its ability to generalize [67, 68, 69, 70, 71]. In the human alignment literature, a similar measure named expressed dimensionality has been studied in the context of neural representations.However, diverging from Ansuini et al. [66], one study found a negative correlation between alignment and this measure [72], and another study found no link [9]. We used the TwoNN method proposed by Facco et al. [73] to estimate intrinsic dimensionality, which makes use of the nearest neighbour distances. First, we linearly scaled all the features to be between 0 and 1. We then computed pairwise distances

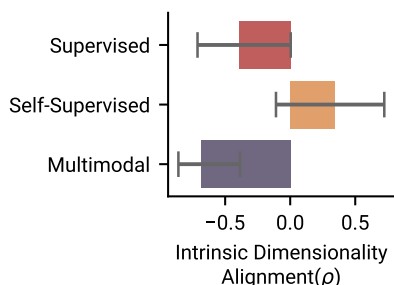

Figure 5: Lower intrinsic dimensionality is linked with higher alignment most strongly for the multimodal models, and to a lesser extent with supervised ones.

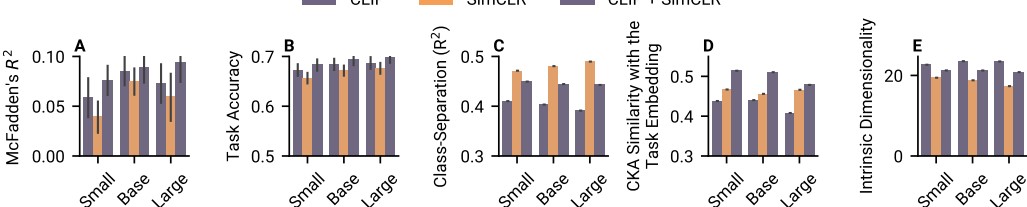

Figure 6: The effect of CLIP loss while controlling for model size and data. We observed that CLIP loss increases alignment when data size and architecture are controlled. Here plotted are (A) McFadden's $R^2$, (B) task accuracy, (C) class-separation, (D) similarity with the task embedding, and (E) intrinsic dimensionality across model sizes and loss functions.

for each pair of data points. Then, we calculated $\mu_i = r_2/r_1$ where $r_1$ and $r_2$ are the shortest distances from datapoint $i$. Later, the empirical cumulative distribution $F^{emp}(\mu)$ was computed by sorting $\mu_i$ and normalising by the total data points $N$. The slope of a linear model that maps $\log \mu_i$ to $-\log 1 - F^{emp}(\mu_i)$ with no intercept gives the intrinsic dimensionality measure.

Pooling over all model types, lower intrinsic dimensionality was significantly associated with alignment ($\rho = -0.32$, $p = .03$, Fig. 4F). However, we found that this relationship was most strongly driven by the multimodal models and to a lesser extent by supervised models (Fig. 5). That input compression and dataset size are positively related to alignment most strongly for CLIP models suggests that the contrastive multimodal training regime unlocks desirable scaling properties in these models. See Fig. 12 in Appendix C for pairwise correlations between the investigated factors.

**Are CLIP models well aligned only due to their high data diet?**

While we found that models trained with contrastive language image loss predicted human behaviour the best, there remains an important confound. These models are also the ones that are trained on the largest datasets (400M to 2B images). Therefore the direct benefits of multimodal training remain unclear. To address this point, we turned to models provided by Mu et al. [52]. Here, the same models are trained on a large dataset (YFCC15M [74, 45]) using three different losses: i) a CLIP loss that penalises for the distance between corresponding pairs of text and image representations ii) a SimCLR [39] loss that pushes the representations of the augmented and the original image close to each other and away from others, and iii) a CLIP + SimCLR loss.

First, we found that CLIP models always fit human data better than SimCLR models, and CLIP + SimCLR models made the best predictions when controlling for model size (Fig. 6A). This suggests that the advantage provided by the CLIP models cannot solely be attributed to the training data. We found the same ranking of models in terms of how well they did the tasks (Fig. 6B). Yet, contrary to our expectations, the SimCLR models had a higher class separation than CLIP models (Fig. 6C), as well as better alignment with the generative task features (Fig. 6D), and lower intrinsic dimensionality (Fig. 6E). This was surprising because, in our previous analyses, we found these properties to be associated with models that predicted human choice better. However, there still may be other confounds that impacted the findings. For example, controlling for training data is not straightforward, as text-image pairs may carry more information than augmented versions of the same image, providing an unfair advantage to the multimodal models.

**Do alignment methods transfer to our learning tasks?**

Lastly, we evaluated the performance of models that were explicitly aligned to be more human-like. This comparison included three sets of models. Fel et al. [11] have aligned models through a method called Harmonization. In addition to the standard supervised training, the models are trained to use the same visual features of images that humans use. The second part is achieved by aligning the networks' saliency maps with feature importance maps obtained from human judgment. This results in networks that perform better in ImageNet and that are aligned with humans. Next, Fu et al. [12] have curated human similarity judgements on a carefully created synthetic dataset. They later fine-tuned pretrained models such as CLIP using Low-Rank Adaptation [75] to derive a metric named DreamSim that outperforms other models in predicting human similarity judgements. Lastly, Muttenthaler et al.

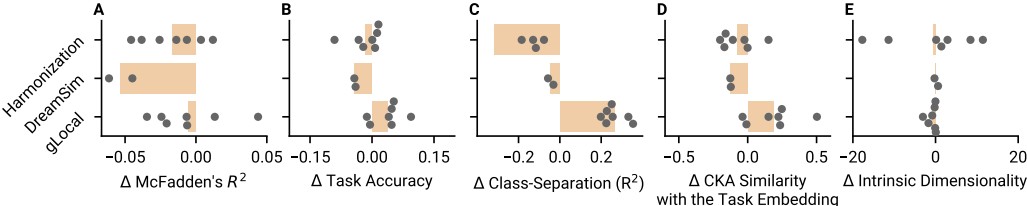

Figure 7: We compared models aligned to humans through three different methods against baselines that had the same architecture and that were pretrained on the same data. We found no consistent improvement in human alignment. Here plotted are (*A*) McFadden's $R^2$, (*B*) task accuracy, (*C*) class-separation, (*D*) similarity with the task embedding, and (*D*) intrinsic dimensionality across model sizes and loss functions.

[10] have fine-tuned representations of pretrained models through a novel transformation named gLocal, which aligns the global representational space to be more human-like by trying to predict human similarity judgements, while preserving the local structure through a contrastive loss that encourages the representations to stay close to their original positions. For these comparisons, we used the models openly provided by the authors.

First, we found that none of the alignment methods improved alignment in our task consistently, with some instances of Harmonised and gLocal models improving alignment (Fig. 7A). Alignment improved task accuracy on average for Harmonised and gLocal models (Fig. 7B). Class separation was lower for all Harmonization and DreamSim models, whereas it increased for all gLocal models tested (Fig. 7C). We also observed that the similarity between the representations and the task embedding decreased after alignment for Harmonization and DreamSim, but it increased in most of the models after gLocal alignment (Fig. 7D). Lastly, we observed heterogeneous patterns in the change of intrinsic dimensionality across the three alignment methods, with gLocal reducing the intrinsic dimensionality for all but one of the tested models (Fig. 7E).

**How do our tasks compare to other alignment measures?**

Lastly, to better characterise how our cognitive tasks fit in the alignment literature, we compared them to previously established measures (Fig. 8). We found the strongest correlation with the THINGS odd-one-out judgements [3, 27] ($\rho = 0.54$ for zero-shot, and $\rho = 0.61$ for probing). Given the two tasks use the same images, and the ground truth of our tasks was constructed from the odd-one-out judgements, this strong relationship is expected. However, the correlations are still moderate, indicating important differences across tasks. Comparisons with an independent similarity judgement [23] task showed a weaker correlation ($\rho = 0.35$), and we found no correlation with a fine-grained two-alternative forced choice task [12]. Lastly, we compared alignment in our task to alignment on the ClickMe dataset [76], which was used to build the Harmonization models [11]. We observed a negative correlation ($\rho = -0.48$) here, suggesting that pixel-level alignment and semantically bound global image alignment might be at odds.

## 5   Discussion

In this work, we investigated the alignment of neural network representations to humans. To study this, we measured how well different neural network representations predict human choices in two newly developed learning tasks. Of the 86 tested representations, all but one predicted human choice above chance level. We furthermore identified several important factors for human alignment, such as large model size, training regime, and low intrinsic dimensionality. These results expand on previous work in both human alignment and cognitive modelling. From an alignment perspective, we considered more challenging tasks compared to previous studies. Previous work has predominantly focused on simple image exposure and similarity judgments. We believe our findings complement this research by addressing unexplored aspects of alignment, which are generalisation and information integration across an extended horizon. From a cognitive modelling perspective, we demonstrated that off-the-shelf pretrained neural networks can serve as representations for cognitive models [77], which allows to push cognitive models into more naturalistic domains.

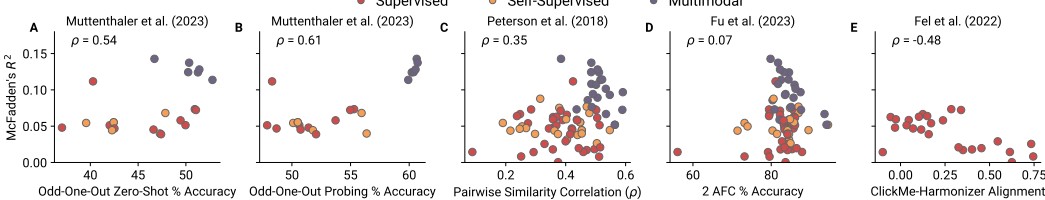

Figure 8: How do our tasks compare to other alignment methods? Our tasks offer similar (but not identical) results with two of the four similarity judgement tasks. There is a strong negative relationship with the ClickMe dataset, which focuses on localised pixel-level alignment.

## 5.1 Related work

How do our findings compare to previous work on alignment? First, previous research has shown that CLIP representations align well with human representations using brain imaging [78, 9, 79] and similarity judgments [3]. In line with this, we also found that the contrastive language-image loss improved alignment when controlling for data size and architecture. We furthermore found that training on large datasets generally improved alignment. Yet, it remains unclear whether supervised training on massive datasets alone can achieve high alignment similar to that of CLIP models. For example, Muttenthaler et al. [3] showed that models trained in a supervised manner on the JFT-3B dataset [61] can outperform CLIP in predicting human similarity judgments. However, since this is a proprietary dataset, we could not make this comparison.

We observed some findings that diverge from previous studies using different experimental paradigms. Muttenthaler et al. [3] and Conwell et al. [9] found no consistent correlation between a model's number of parameters and its alignment with human similarity judgments and visual cortex activity. In contrast, we found that models with more parameters were more predictive of human choices. Another significant divergence is how intrinsic dimensionality relates to alignment. Elmoznino and Bonner [72] found that vision models with higher latent dimensionality better predict visual cortex activity, Conwell et al. [9] found no correlation. In contrast to this, we observed that lower intrinsic dimensionality led to increased alignment for CLIP models. We hypothesize that both of the observed discrepancies are due to the higher cognitive demands required by our tasks, highlighting the importance of studying alignment in more complex settings. That being said, an alternative explanation for the latter discrepancy could be due to differences in measuring latent dimensionality. Both Elmoznino and Bonner [72] and Conwell et al. [9] use the squared sum of the eigenvalues of principal components divided by the sum of squares of eigenvalues, assuming representations lie on a linear manifold. However, previous work shows that later layers in vision models lie on a curved manifold [66]. Thus, using principal components might not be the best method for this estimation.

Lastly, we found that a method designed for increasing human alignment, DreamSim [12], actually hurt alignment in our task. On the other hand, gLocal [10] and Harmonization [11] improved both performance and human alignment for some models but not all of them. However, the gLocal transform heavily utilises the THINGS dataset, as it made use of the triplet odd-one-out similarity judgement data [27], making it difficult to interpret how well it generalises to other settings. Taken together, these results highlight the importance of studying how well different alignment methods transfer across tasks, as we have done in this work.

## 5.2 Limitations

There are several limitations and extensions of our work that deserve to be highlighted. The main limitation concerns the interactions between factors in the tested neural networks, making it difficult to isolate specific factors. For example, we would like to test the influence of loss function keeping all other factors equal. Ideally, we would train all combinations of architectures, model sizes, loss functions, and datasets, but this is computationally infeasible.

While we controlled for factors such as training data size and architecture in our comparison of CLIP to other models, there may still be confounding variables we haven't accounted for. For instance, it's not straightforward to compare the information content of image-text pairs used in CLIP training to

image-only data used in other models. Text-image pairs might inherently carry more information than single images, potentially giving multimodal models an advantage that's difficult to quantify. This and other subtle differences in training paradigms could influence our results in ways that are challenging to isolate and measure. Lastly, there can also be other families of models that may outperform CLIP models we haven't considered, such as video models, generative models, or image segmentation models.

We furthermore tested only two experimental paradigms. Future research should explore whether the considered representations predict human behaviour with nonlinear task rules and extend to other paradigms. In particular, one should also consider tasks beyond those generated through the embedding from Hebart et al. [27]. Previous work has shown that it is possible to automatically generate a large set of text-based category learning problems using large language models [80]. It might be interesting to test whether these methods can be extended to generate tasks involving visual stimuli and use these tasks to test whether our findings generalise to a wider setting.

Finally, we only measured human alignment by looking at behaviour. However, to fully confirm our results, it would also be important to investigate the alignment to brain data. Hence, future work should replicate our experiments in an MRI scanner and compare the representations of neural networks to people's brain activity.

### 5.3 Conclusion

The findings presented in this work have implications both for machine learning and cognitive science. For machine learning, our task and modelling approach offers a new way to measure the human alignment of neural network representations and use this as a metric while building human-aligned neural networks. Alignment at this level can pave the way for artificial systems that can generalise across semantically rich tasks, making them more robust and powerful. For cognitive science, our findings create the opportunity to study several other problems in naturalistic settings by showing that people can do learning tasks with naturalistic stimuli and that pretrained neural networks can be used to extract representations for cognitive models. This could open up the door for a whole new cognitive science that uses naturalistic tasks and environments and thereby increase the validity of the cognitive sciences more generally.

## Acknowledgments and Disclosure of Funding

This work was funded by the Max Planck Society, the Volkswagen Foundation, as well as the Deutsche Forschungsgemeinschaft (DFG, German Research Foundation) under Germany's Excellence Strategy–EXC2064/1–390727645.

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

# A    Methods

**Participants** For the category learning task, we recruited 98 participants (48 females, 50 males, mean age= 28.92y, SD= 7.32) on the Prolific platform. Participants with less than $50\%$ accuracy were excluded from the analyses, leaving us with 91 participants. A base payment of £ 1.50 was made, and participants could earn an additional bonus of £ 6.00. The median completion time was 12 minutes and 38 seconds. The inclusion criteria included having a minimum approval rate of $97\%$, and a minimum number of 15 previous submissions on Prolific. Participation in the reward learning study was an exclusion criterion. For the reward learning task, 99 participants were recruited (49 females, 49 males, 1 other, mean age = 27.9 y, SD = 9.13). After applying the $50\%$ accuracy criteria, we were left with 82 participants. A base payment of £ 2.00 was made, and an additional performance-dependent bonus of £4.00 was offered. The median completion time was 9 minutes and 26 seconds. The inclusion criteria included having a minimum approval rate of $95\%$, and a minimum number of 10 previous submissions on Prolific. All participants agreed to their anonymized data being used for research. The study was approved by the ethics committee of of the medical faculty of the University of Tübingen (number 701/2020BO). Participants gave consent for their data to be anonymously analyzed by agreeing to a data protection sheet approved by the data protection officer of the MPG (Datenschutzbeauftragte der MPG, Max-Planck-Gesellschaft zur Förderung der Wissenschaften).

**Tasks and Stimuli** Both tasks were run online in forced full-screen mode. Participants were shown written instructions and were asked to complete comprehension check questions before they could start the tasks. In both tasks, participants were given unlimited time to make decisions. In the category learning task, binary (correct versus wrong) feedback was given for $2s$. In the reward learning task, the associated reward with the stimuli was shown for $1.5s$, and there was an inter-trial interval of $1s$ where participants were shown a blank screen. Throughout both tasks, the estimated total payment of participants was shown on the upper part of the screen. At the end of the tasks, participants were asked whether they thought their data should be used for analysis. Across both tasks, all but one participant responded saying their data should be analyzed, whose data was anyway excluded due to poor performance. The category learning task was programmed using jsPsych [81], whereas the reward learning task was programmed in plain JavaScript.

For each participant, 120 stimuli were sampled independently from the THINGS database. Because the loadings of the features were not uniformly distributed, we made 5 equally sized bins of the loadings for the assigned feature and sampled object categories uniformly from these bins. From these object categories, the specific images were assigned randomly. For details on the used features and the embedding, see Hebart et al. [27].

**Behavioural Analyses** We used mixed-effects logistic models for both category and reward learning analyses. For category learning, we predicted correct responses per trial, using trial number as a fixed effect and including participant-specific random effects for intercept, trial number, and assigned task rule. In the reward learning model, we predicted whether the image on the right is selected, incorporating the trial number, reward difference between images, and their interaction as fixed effects. These factors, along with the assigned task rule, were also modelled as participant-specific random effects. Both models effectively captured task structure, learning progression, and individual variability in performance. In R formula notation[82, 83], the model for category learning is denoted as follows:

correct_choice $\sim$1 + trial + (1 + trial + dimension | participant)

For the reward learning task, the following model was used:

right_choice $\sim$−1 + trial $*$ right_left_reward_difference +
(−1 + trial + dimension + right_left_reward_difference | participant)

where $-1$ denotes no intercept.

**Software, Data, & Compute Resources** The code to reproduce the reported results is available at https://github.com/candemircan/naturalcogsci and we provide anonymised human choice data on https://osf.io/h3t52/. For the learning models we used lme4 [83] and scikit-learn [84]. To extract representations from neural networks we used thingsvision [78].

Computations were performed on an academic SLURM cluster. Feature extraction was done on a single Nvidia A100 GPU (40GB) under 24 hours. The linear models were parallelised across several jobs that used single core and 8GB RAM and were completed in under 48 hours. The mixed-effects models were similarly parallelised and completed under 24 hours.

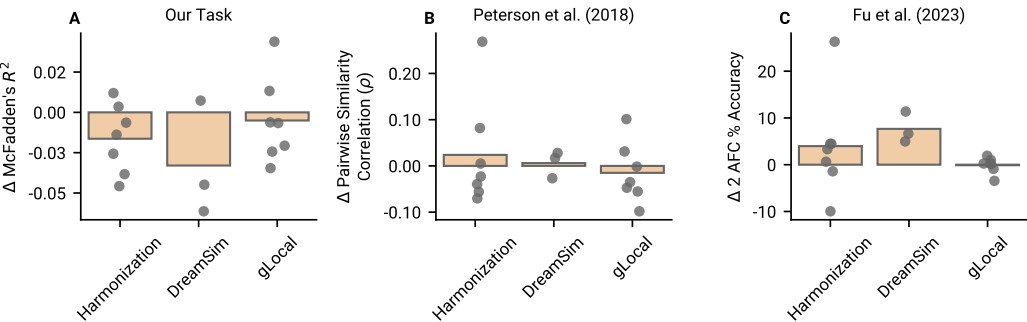

Figure 9: Change of human alignment for different methods on different datasets

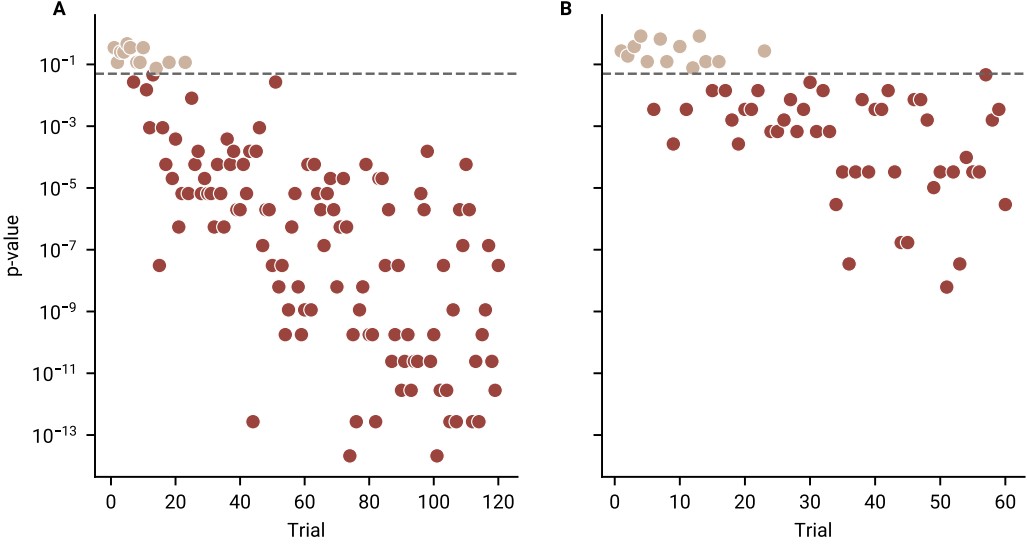

Figure 10: Participant Performance Against Chance Level at Each Trial. Trial-by-trial p-values from 1 sample t-tests testing accuracy against chance level for (*A*) category learning task and the (*B*) reward learning task.

## Modelling

For the category learning task, we used an $\ell_2$ regularised logistic regression model to optimize regression weights. We relied on `scikit-learn`'s `LogisticRegression` class which internally optimizes the following objective:

$$\mathbf{w}^* = \arg\max_{\mathbf{w}} \sum_{i=1}^{N} -c_i \log(p(c_i|\mathbf{x}_i, \mathbf{w})) - (1 - c_i) \log(1 - \log(p(c_i|\mathbf{x}_i, \mathbf{w})) + \frac{1}{2}||\mathbf{w}||_2^2 \quad (5)$$

For the reward learning task, we used a Bayesian linear regression model to infer a posterior distribution over regression weights. We relied on `scikit-learn`'s `BayesianRidge` class which infers a posterior distribution assuming spherical Gaussian priors (i.e., $p(\mathbf{w}) = \mathcal{N}(0, \lambda^{-1}\mathbf{I})$) and Gaussian likelihood (i.e., $p(y_i|\mathbf{x}_i, \mathbf{w}) = \mathcal{N}(\mathbf{w}^\top \mathbf{x}_i, \beta^{-1})$). Based on these assumptions, the posterior

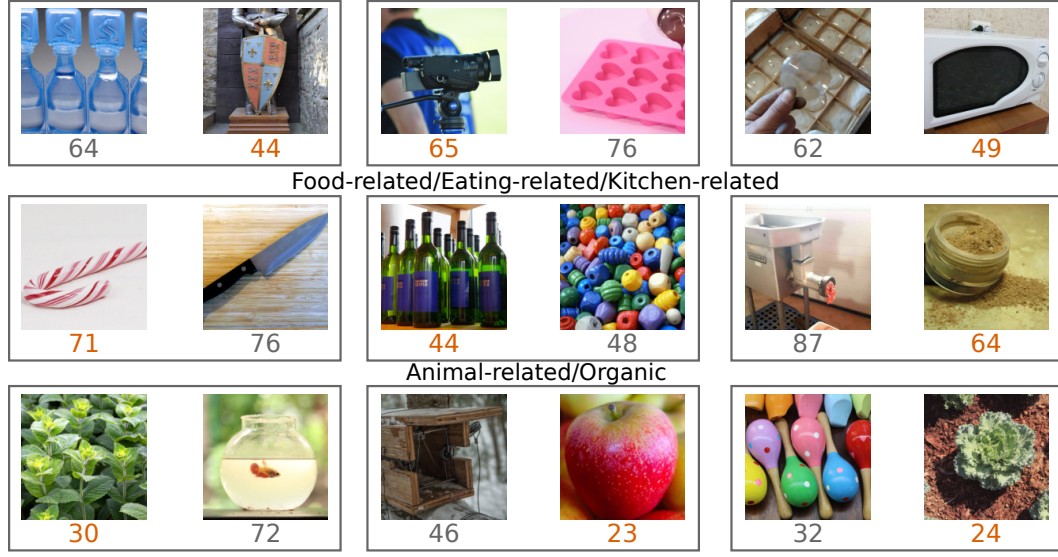

Figure 11: Example trials showing the similarity between CLIP and human decisions that show disagreement with the task embedding. Each row shows three trials from a different condition. Orange highlighted text shows the option chosen by all CLIP models and the human participant, whereas grey text shows the decision made by the task embedding. As the tasks were generated using the task embedding, all the choices shown here made by CLIP and humans are suboptimal. Shown examples are from the second half of the task, as to eliminate the learning process as a confound. The original images are replaced with copyright-free alternatives from the THINGSplus database [85].

distribution can be computed in closed form:

$$p(\mathbf{w}|\mathbf{X}, \mathbf{y}) = \mathcal{N}(\mathbf{m}_N, \mathbf{S}_N) \tag{6}$$

$$\mathbf{m}_N = \beta \mathbf{S}_N \mathbf{X}^\top \mathbf{y} \tag{7}$$

$$\mathbf{S}_N^{-1} = \lambda \mathbf{I} + \beta \mathbf{X}^\top \mathbf{X} \tag{8}$$

where $\mathbf{X}$ and $\mathbf{y}$ denote the stacked inputs and targets respectively.

We run both models from scratch on each trial using all previously observed input-target pairs. The choice of these models was motivated by previous investigations in similar – but low-dimensional – settings [86, 87, 88].

The $\ell_2$ penalty term for the logistic regression model described above was determined via grid search to maximise task performance, on a per participant basis. For the linear regression model, $\lambda$ and $\beta$ were fitted to maximise the log marginal likelihood on the task performances.

The estimates from these models were used in mixed-effects logistic regression models with leave-one-out predictions to assess participant choices. For category learning, we used logistic regression probability estimates as predictors. In the reward learning task, we used the difference in estimated rewards from linear regression models as predictors. In both cases, these predictors were included as both fixed and random effects, allowing us to account for individual differences while maintaining the group effects. These correspond to the following models in R formula notation for category and reward learning respectively:

human_choice $\sim -1$ + probability_estimate + ($-1$ + probability_estimate | participant)

human_choice $\sim -1$ + estimated_reward_difference + ($-1$ + estimated_reward_difference | participant)

For the mixed-effects models, the training data was centred and divided by its standard deviation. The same scaling parameters were applied to the test data.

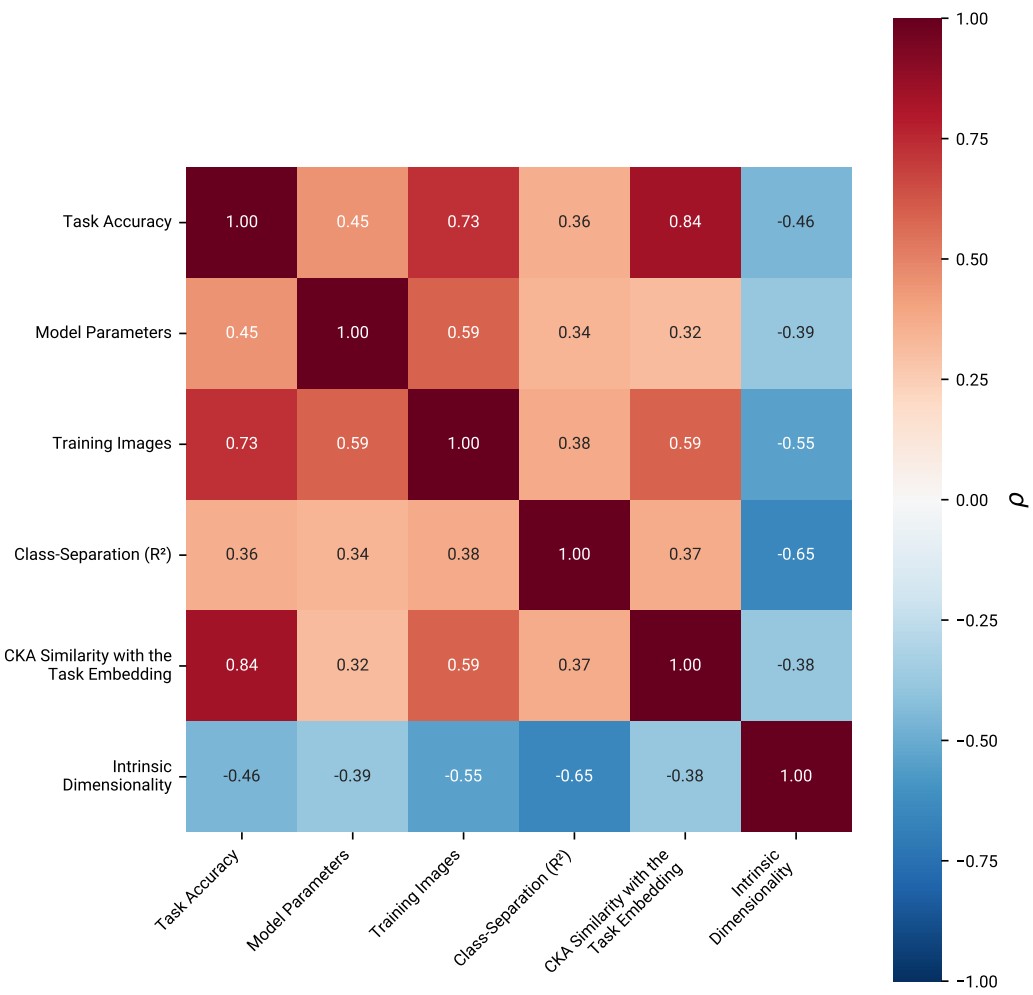

Figure 12: Pairwise Spearman correlations between the factors investigated that contribute to alignment.

**Additional human-alignment tasks**

Results reported in Fig. 8A & Fig. 8B include comparisons for the overlap of models between those reported by Muttenthaler et al. [3] and the ones we tested. For Fig. 8B & Fig. 8C, we tested all the vision models reported in our paper. However, for the Peterson et al. [23] dataset, we only found a subset of the original data reported in the paper[1]. The ClickMe-Harmonizer alignment was only computed for supervised models, as the method requires computing gradients for ImageNet classes, which we could only do for the supervised models that had ImageNet classification heads.

## B    Testing aligned models on other datasets

Above, we also report how different alignment methods perform on different datasets (Fig. 9). Harmonization is on average slightly more human-like on the two external tested datasets compared to baselines. DreamSim shows mixed results for the Peterson et al. [23] dataset, but it shows improvement on the NIGHTS dataset [12]. This is not surprising, as this dataset was used to build DreamSim. Lastly, gLocal shows mixed results.

---

[1]Specifically, we tested similarity judgements obtained on Animal, Fruit, and Vegetable categories. The data was obtained from [89]

# C   Additional results

Above we provide some additional results supporting our claims in the main text. Fig. 10 shows participants can do both tasks above chance level very early on in the task. Fig. 11 shows some incorrect choices made by humans and also by CLIP models, and Fig. 12 shows pairwise correlations between the factors we investigated that contribute to alignment.

