# OpenReview forum: "Evaluating alignment between humans and neural network representations in image-based learning tasks"
_NeurIPS.cc/2024/Conference — NeurIPS 2024 poster_

### Official Review · Reviewer_PLVt · 2024-07-02

**Soundness:** 2
**Presentation:** 2
**Contribution:** 3
**Rating:** 5
**Confidence:** 3

**Summary:**

This article assesses how AI embedding can predict human behavior on 2 different tasks. In those tasks, humans have to learn continuous relationships and categories of natural images. Overall the authors tested the predictability of 77 retrained neural networks, including supervised, semi-supervised, and multimodal models. The authors found that a larger training size, a higher number of parameters, and lower intrinsic dimensionality improve the human alignment of the tested model. In addition, the authors claim that multimodal networks are better aligned with humans. In the last part, the authors test 3 different alignment methods and show that only the gLocal method shows a better predictability of human performance on the considered task.

**Strengths:**

This article introduces a novel behavioral benchmark to compare humans and machines. This benchmark (provided that the data are public) should be a useful asset in the human-machine comparison toolbox. The number of tested models is both large and also diverse.

**Weaknesses:**

I found 3 weaknesses:
* 1 - Given the data presented in the article, I have the feeling that claiming that ‘multimodality better predicts human performance’ is a bit overstated. It seems to have plenty of confound factors (besides the training side) that haven’t been explored and that might prove that this claim is not entirely true
* 2 - Some of the parts in the article are very unclear with several details missing. I have the feeling I did not have clear enough to judge the methods used by the authors (which is problematic). Please refer to the question section for more details.
* 3 - The last analysis (on the alignment method) is pretty weak compared to the rest of the article (not enough models and not enough model diversity, no statistical analysis…).

Note that I am willing to increase my rating if the questions are properly addressed.

**Questions:**

* Q1: The section 3 is not clear enough. For example (line 95): « Using an intercept and the trial number ». What does it mean? To my (maybe not sufficient) knowledge, the intercept is the constant term in a linear fit. If this is the case, what kind of data are you using to regress to find the fit? And most importantly, how does this intercept relate to human performance?

* Q2: Still in section 3, the results of the statistical tests are poorly explained. For example (line 98): what does (Beta, z) relate to? Is that the intercept and the number of trials? Is that something else? This should be explicitly stated and should be put under the carpet!

* Q3: In section 4, there is not enough detail on how you train the linear probe to predict the performance of the category learning and the reward learning task. Especially in the case of the ‘linear regression mode with spherical prior » (line 121). What does it mean, what is the loss you exactly optimize to find the regression parameters? Why Gaussian prior? This should be motivated and better explained...

* Q4: Even more dark for me is line 171. T is the generative task representation. What does it mean, and how this is obtained? What is the generative task? What’s the experimental protocol for this task (if there is one)…

* Q5: I am lost in Figure 5. If I am not wrong clip is represented by the yellow color (multimodal, as confirmed in Fig 3). But in the caption of the same figure you say: « lower intrinsic dimensionality only increases alignment for CLIP models ». But the supervised model (red bar) showcases a similar trend. Is that a typo? Do I miss something here?

* Q6: In Figures 5, 6, there is no error bar and statistical test to assess if the shown results are significant. For example in Fig6, the differences between CLIP, SimCLR, and CLIP+SimCLR seem to be rather small and I cannot evaluate if those differences are significant.

* Q7: In Figure 3, would it be possible to include inter-human reliability (i.e. how much different participants are aligned with each other)

* Q8: Is it not clear how to control the number of training data in CLIP and SimCLR? In CLIP, this is pairs of data (language/image) whereas SimCLR is only on images (and it’s an augmented version, but it might not count as it is not external information per say).

* Q9: The authors suggest that the training size is a confound (and I agree), but the model’s number of parameters might also be a confound (that might be related to the training size). This is a problem because most of the multimodal models have generally a high number of parameters (see Fig B). How do I know if this is multimodality is the number of parameters that increase the human alignment? Would it be possible to run a similar experiment as in Fig 6, but testing the number of parameters?

* Q10: I have the feeling there is no enough data (and a too small variety of tested model) to back the claim : « We found that only one alignement method improved predictive accuracy » (line 15). For example, for the harmonization technique only supervised models have been tested, and several other models have been tested with gLocal. How do you know that « harmonising » other type of model do not improve the predictability on your task. And I am even sure that the tested architectures are the same for the supervised harmonized and supervised gLocal models ? How do you want me to be convinced if on top of the alignement methods, other parameters can vary (as architecture…). This analysis clearly lack rigor...

**Limitations:**

The authors properly discuss the limitation of their work

---

> ### Author Rebuttal · Authors · 2024-08-06
>
> > (...) claiming that 'multimodality better predicts human performance' is a bit overstated.
>
> We agree that many factors contribute to alignment, as detailed in Figure 4. However, the comparison of SLIP, SimCLR, and CLIP models in Figure 6 suggests image language training is important, as architecture and dataset are matched and downstream performance is comparable [1]. We acknowledge there may be unexplored factors and welcome suggestions for additional comparisons.
>
> > Q1: The section 3 is not clear enough.
>
> For category learning, we used a mixed-effects logistic model to predict correct responses per trial. Main effects included intercept and trial number, where the intercept models above chance performance and the trial number models learning. For reward learning, we predicted which image was chosen using a similar model with trial number, reward difference, and their interaction as main effects. Here, the interaction captures the learning effect. The same variables were also used as random effects for individual differences.
>
> > Q2: (...) For example (line 98): what does (Beta, z) relate to? Is that the intercept and the number of trials?
>
> The first set of numbers are for the intercept and the second for the trial number. $\hat{\beta}$ is the estimated regression coefficient. z is the test statistic comparing predictor coefficients against their standard errors to test the null hypothesis that the predictor’s coefficient is 0.
>
> > Q3: In section 4, there is not enough detail on how you train the linear probe
>
> We added the following details to the Appendix:
>
> **"For the category learning task, we used an  $\ell_2$ regularised logistic regression model. We relied on `scikit-learn`’s `LogisticRegression` class which optimizes the following objective:**
>
>
> $$\mathbf{w}^* = \underset{\mathbf{w}}{argmax} \sum_{i=1}^N -c_i \log(p(c_i | \mathbf{x}_i, \mathbf{w})) - (1-c_i) \log(1-\log(p(c_i | \mathbf{x}_i, \mathbf{w})) + \dfrac{1}{2} ||\mathbf{w} ||^2_2$$
>
> **For the reward learning task, we used a Bayesian linear regression model to infer a posterior distribution over regression weights. We relied on `scikit-learn`’s `BayesianRidge` class which infers a posterior distribution assuming spherical Gaussian priors (i.e., $p(\mathbf{w}) = \mathcal{N}(0, \lambda^{-1}\mathbf{I})$) and Gaussian likelihood (i.e., $p(y_i | \mathbf{x}_i, \mathbf{w}) = \mathcal{N}(\mathbf{w}^{\top}\mathbf{x}_i, \beta^{-1})$). Here, posterior distribution can be computed in closed form:**
>
> $$p(\mathbf{w} | \mathbf{X}, \mathbf{y}) = \mathcal{N}(\mathbf{m}_N, \mathbf{S}_N)$$
> $$\mathbf{m}_N = \beta \mathbf{S}_N \mathbf{X}^{\top} \mathbf{y}$$
> $$\mathbf{S}^{-1}_N =  \lambda \mathbf{I} + \beta \mathbf{X}^{\top}\mathbf{X}$$
>
> **where $\mathbf{X}$ and $\mathbf{y}$ denote the stacked inputs and targets respectively.**
>
> **We run both models from scratch on each trial using all previously observed input-target pairs. The choice of these models was motivated by previous investigations in similar but low-dimensional settings [2,3,4].“**
>
> > Q4: What is the generative task?
>
> T is the ground truth of the task, obtained by [5]. They trained a low-dimensional sparse embedding model on the odd-one-out similarity judgements on THINGS. For both tasks, participants were assigned to conditions defined over one of the top three embedding dimensions (Figure 1C). Please see lines 72-91 for a detailed account.
>
> > Q6: In Figures 5, 6, there is no error bar and statistical test to assess if the shown results are significant.
>
> Please see Rebuttal Figures 5 & 6. In Figure 5, we only report the intrinsic dimensionality analysis because the rest are not significant, which we earlier pointed in Appendix Table 3. We hope this also addresses Q5. For Figure 6, we found CLIP $R^2$ values were higher than SimCLR $R^2$ values ($\hat{\beta} = .014, t=1.98, p=.048$), and SimCLR + CLIP $R^2$ values were higher than CLIP $R^2$ values ($\hat{\beta} = .03, t=3.91, p<.0001$).
>
> > Q7: In Figure 3, would it be possible to include inter-human reliability
>
> As the sequences of observations were randomly assigned and not identical, inter-rater reliability cannot be calculated.
>
> > Q8: Is it not clear how to control the number of training data in CLIP and SimCLR?
>
> We'll add: **"Comparing image-language pretraining with augmented image training remains challenging, as language may provide information beyond an augmented version of the same image."**
>
> > Q9: the model's number of parameters might also be a confound
>
> Model size is important (Figure 4B), but some smaller CLIP models outperform larger ones, and some large self-supervised models are outperformed by smaller CLIP models. Appendix Table 3 shows no relationship between size and score for CLIP models, suggesting performance isn't solely driven by model size.
>
> > Q10:I have the feeling there is no enough data (and a too small variety of tested model) to back the claim : « We found that only one alignment method improved predictive accuracy » (line 15).
>
> We  added more baseline comparisons and tested alignment methods on two additional datasets (Rebuttal Figure 1 & 2). DreamSim consistently improves alignment in its test set, gLocal shows the largest improvements in our task, while Harmonization degrades alignment. We are limited to the models trained by the original authors, which allows us to only offer a qualitative report. Harmonization alignment is only possible for supervised models by the nature of the method.
>
> [1] SLIP: Self-supervision meets Language-Image Pre-training, Mu et al., arxiv 2021
>
> [2] Learning strategies in amnesia, Speekenbrink et al., Neuroscience & Biobehavioral Reviews 2008
>
> [3] A unifying probabilistic view of associative learning, Gershman, PLOS CB 2015
>
> [4] Heuristics from bounded meta-learned inference, Binz et al., Psychological Review 2022
>
> [5] Revealing the multidimensional mental representations of natural objects underlying human similarity judgements. Hebart et al., Nat. Hum. Beh. 2020

---

> > ### Comment · Reviewer_PLVt · 2024-08-12
> > **response**
> >
> > I still think the claim of this article is overstated, and even if results tend to show an interesting trend, there is still many confound that has not been explored. I would have appreciate to downplay the overclaim (and to discuss more the other possible confound). I maintain my rating.

---

> > > ### Author Response · Authors · 2024-08-12
> > > **Softening the Claims About Multimodality**
> > >
> > > Thank you for this helpful feedback. We will update our manuscript to soften the claims about multimodality in two regards:
> > >
> > > 1. **We emphasise that these claims are concerning the models we tested and those that are currently publicly available.**
> > > 2. **We further discuss how additional factors that we did not consider can affect alignment.**
> > >
> > >
> > > Regarding the 1st point, we made the following changes:
> > >
> > > - In the Abstract (Line 9): We found that while training dataset size was a core determinant of alignment with human choices, contrastive training with multi-modal data (text and imagery) was a common feature of **currently publicly available models** that predicted human generalisation.
> > >
> > > - In the Introduction (Line 52): While almost all **tested** models generalised above chance level and predicted human behaviour in both naturalistic learning tasks, contrastive language image pretraining (CLIP) [45] consistently yielded the best predictions of human behaviour **out of the models we tested.\footnote{While our analysis already considered an extensive set of models, we want to emphasise that it only presents a snapshot of the current model landscape, and that our results are therefore subject to further validation once new models become available.}**
> > >
> > > And for the 2nd point, we added the following new paragraphs to the text:
> > >
> > > - In the Introduction (Line 55): **However, it's important to note that this observation may be influenced by various factors beyond just the training approach. While our analysis suggests that the training diet alone does not fully account for CLIP's performance, there could be other contributing elements that require further investigation.**
> > > - In the Results (Line 213): **However, there still may be other confounds that impacted the findings. For example, controlling for training data is not straightforward, as text-image pairs may carry more information than augmented versions of the same image, providing an unfair advantage to the multimodal models.**
> > > - In the Limitations (Line 290): **While we controlled for factors such as training data size and architecture in our comparison of CLIP to other models, there may still be confounding variables we haven't accounted for. For instance, it's not straightforward to compare the information content of image-text pairs used in CLIP training to image-only data used in other models. Text-image pairs might inherently carry more information than single images, potentially giving multimodal models an advantage that's difficult to quantify. This and other subtle differences in training paradigms could influence our results in ways that are challenging to isolate and measure. Lastly, there can also be other families of models that may outperform CLIP models we haven’t considered, such as video models, generative models, or image segmentation models.**

---

### Official Review · Reviewer_MCn8 · 2024-07-10

**Soundness:** 3
**Presentation:** 3
**Contribution:** 3
**Rating:** 6
**Confidence:** 3

**Summary:**

This paper explores the alignment between human cognitive processes and neural network representations in image-based learning tasks. The authors evaluated 77 pretrained neural network models to see how their representations aligned with human learning patterns in two tasks: category learning and reward learning.

They found that models with contrastive training, especially CLIP models, effectively predicted human generalization. Factors such as training dataset size, model size, and representation properties were identified as influential for alignment with human cognition.

**Strengths:**

The paper makes substantial contributions by:
1. Introducing two novel tasks to evaluate human-model alignment.
2. Demonstrating that larger contrastive training datasets and multi-modal data improve predictions of human behavior.
3. Conducting extensive and engaging human experiments.

Additionally, I found that:
- Figure 4 is particularly interesting and stimulates a lot of valuable discussion.
- It would have been very insightful to include a comparison with generative models (e.g., EBM, VAE) and relate these findings to the brain's generative hypothesis.

**Weaknesses:**

Despite its strengths, the paper has several weaknesses. These are categorized into major problems (**M**) and minor problems (_m_).

**M1**: There is a potential bias in using the THINGS database for both training and evaluation, especially with the gLocal models, which might lead to circular reasoning. The space trained with gLocal is implicitly designed to perform well on these specific tasks, which could inflate the perceived performance of the models. Addressing this concern by diversifying the evaluation datasets or providing a more detailed justification for using THINGS would strengthen the paper. You could use a measure from DreamSim and Harmonization as control.

_m1_: The use of $\ell_2$ regularization in the regression model may penalize some tokens disproportionately due to their high activation values. Standardizing all activations before performing regression could mitigate this issue and provide a fairer comparison.

_m2_: Concerning the use of the TwoNN method for estimating intrinsic dimensionality, could you verify that standardization instead of linear scaling yield the same results, as we know that some high norm token (outlier token) exist and could totally distort your distances.

_m3_: Figure 7 lacks clarity on which models are used in the harmonization method? You have 7 Harmonization model in Fig. 3 and only 3 in Fig. 7. Or at least specify how they were chosen.

_m4_: Could you also add a metric to complement McFadden (e.g., avg accuracy) ?

**Questions:**

See Strengths & Weaknesses

**Limitations:**

Yes, the limitations identified by the authors are accurate and well-documented. Regarding the weakness I mentioned, I reserve the right to increase the score if the authors adequately address my major concerns.

---

> ### Author Rebuttal · Authors · 2024-08-06
>
> > It would have been very insightful to include a comparison with generative models.
>
> Thanks for the great suggestion! We now included 3 masked autoencoder models[1] trained on ImageNet in our comparison. We chose these specific models because they are SOTA classification models trained in a generative manner. They don’t do well in predicting human choice (mean $R^2$ = 0.05, min $R^2$=0.03, max $R^2$=0.06). Therefore, it appears that training generative models doesn't provide a particular advantage, at least in our comparisons. We will include these results in the camera-ready version.
>
> > **M1**: There is a potential bias in using the THINGS database for both training and evaluation, especially with the gLocal models,
>
> Yes, this is an important limitation that we have also pointed out in line 279. Nevertheless, we wanted to include this comparison with that caveat. To our surprise, despite the shared data across the gLocal training and our task,  we did not observe consistent improvements in model fit with the gLocal compared to the baselines.
>
> > Addressing this concern by diversifying the evaluation datasets (...) You could use a measure from DreamSim and Harmonization as control.
>
> Thank you for these suggestions. We have now compared the alignment methods tested across two additional tasks that do not use the THINGS data (Rebuttal Figure 2). Similar to our initial findings, gLocal leads to improved alignment in limited cases. We found that Dreamsim consistently improves alignment on its test set, but no consistent improvements are observed on the independent dataset.
>
> Additionally, we have incorporated four more baseline models for Harmonization and one more for DreamSim. These new baselines led to some modest improvements in our task, although they were not consistent and were smaller than those achieved by the gLocal models. As observed in our task, Harmonization models tend to reduce alignment in most cases. We will include these results in the camera-ready version.
>
> We would also have liked to make this comparison for the Harmonization metric on the Harmonization dataset. However, this was not possible because the Harmonization metric requires a trained Imagenet classification head, which Dreamsim and gLocal models do not have.
>
> > (...) or providing a more detailed  justification for using THINGS would strengthen the paper.
>
> We chose to conduct our experiments with the THINGS data because the ground truth features that we used were only available for THINGS. This is because the ground truth features are a low dimensional embedding that was learned to predict human odd-one-out similarity judgements on the THINGS dataset. This embedding is central to the design of our tasks and unfortunately, it does not generalise to new image data. Therefore, we had to use the THINGS images. We will add this justification to the camera-ready version.
>
> > m1: (…) Standardizing all activations before performing regression could mitigate this issue and provide a fairer comparison.
>
> Thanks for pointing out this important detail. This is indeed what we have done in our comparisons. We scaled the features using the mean and the standard deviation of the training data and applied the same transformation to the test data. We did this both in our mixed-effects models and also the linear and logistic regressions that were used to fit the features to the tasks.
>
> > m2: Concerning the use of the TwoNN method for estimating intrinsic dimensionality, could you verify that standardization instead of linear scaling yield the same results
>
> Thanks for the suggestion. We have now made this comparison including the new models we added and observe that min-max scaling ($\rho$=-0.23, $p$=0.04) and standard scaling ($\rho$=-0.24, $p$=0.03) yield near identical results.
>
> > m3: Figure 7 lacks clarity on which models are used in the harmonization method?
>
> We were only including the Harmonization models for which we had baselines. We have now added baselines for all the Harmonization models and therefore include all Harmonization models in our comparison. The results are highly similar to what was reported in the initial submission.
>
> > m4: Could you also add a metric to complement McFadden (e.g., avg accuracy) ?
>
> Yes, please see Rebuttal Figure 3. The maximum accuracies that are achieved are 68% for the category task and 67% for the reward task. Sorting the models by accuracy and $R^2$ alters the ordering of the models slightly, although the rank correlations are very high both for category ($\rho$=.97) and the reward ($\rho$=.93) tasks and our qualitative results stay the same, where 9 of the top 10 models are CLIP variants in both tasks. We will include the average accuracy in the final version.
>
> [1]He, K., Chen, X., Xie, S., Li, Y., Dollár, P., & Girshick, R. (2021). Masked Autoencoders Are Scalable Vision Learners.

---

> > ### Comment · Reviewer_MCn8 · 2024-08-12
> >
> > Thank you for addressing my concerns and adding the extra experiments.
> >
> > I believe these additions definitely make the paper stronger. While I still have some reservations about the circularity issue with the THINGS database, I appreciate the steps you've taken to address it. I'll be increasing my score to 6.
> >
> > Good luck with acceptance!

---

> > > ### Author Response · Authors · 2024-08-12
> > >
> > > Thanks very much for your response and for raising your score! We are glad you found the new analyses useful.

---

### Official Review · Reviewer_NCLd · 2024-07-10

**Soundness:** 2
**Presentation:** 3
**Contribution:** 3
**Rating:** 6
**Confidence:** 4

**Summary:**

The authors propose a novel representational alignment metric tied to sequential human behavior and leverage it to analyze factors in NN design and training that contribute to increased alignment with humans.

**Strengths:**

- Great analysis of factors contributing to alignment of NNs with humans
- Well-written manuscript, easy to follow
- Great descriptions of human and NN experiments; important for reproducibility
- Important and timely contribution to a growing field (representational alignment)
- I agree with the authors that its important to tie notions of alignment to behavior (a metric is only useful if it actually predicts something downstream)

**Weaknesses:**

- The authors claim to make 2 contributions: a novel alignment metric and an analysis of factors contributing to human alignment of various NN that leverages that alignment metric. The authors claim that their alignment metric is better than existing alignment metrics as it requires "generalisation and information integration across an extended horizon". However, there isn't a comprehensive comparison to previous alignment metrics to see (a) how correlated the new metric is to existing metrics, and (b) whether human behavior on the tasks can be predicted from previous metrics. For (b) for example, the authors could take human pairwise similarity judgments and treat them as features from which to learn the task and then correlate that with human responses (i.e., treating the sim judgments as a kernel). This kind of analysis would go a long way in validating the proposed metric and comparing/contrasting it against existing alignment metrics.
- Perhaps I missed it, but I didn't see an analysis of inter-rater reliability/noise roof. This would be important to put the McFadden R^2 numbers into perspective and understand how good alignment *could* be.
- Happy to raise my score if these points and questions below are resolved

**Questions:**

- why were participants with <50% accuracy excluded?
- why might the alignment numbers be so low? McFadden's R^2 is typically considered to be excellent fit in 0.2-0.4 range, yet all the results cap out at about 0.15

**Limitations:**

- Limitations are adequately discussed (assuming weaknesses and questions raised above are addressed)

---

> ### Author Rebuttal · Authors · 2024-08-06
>
> >  The authors claim that their alignment metric is better than existing alignment metrics as it requires "generalisation and information integration across an extended horizon".
>
> We believe that our method is not necessarily better than other metrics but that alignment is multifaceted. We will clarify this stance further in the main text by updating line 248 as follows:
> **"Previous work has predominantly focused on simple image exposure and similarity judgments. We believe our findings complement this research by addressing unexplored aspects of alignment, which is  generalisation and information integration across an extended horizon."**
>
> > how correlated the new metric is to existing metrics
>
> This is an excellent question that’s crucial for adequately placing our work in the field of alignment. To address this, we compared our alignment metric to four different metrics computed on different datasets. The results are shown in Rebuttal Figure 1. We will add the following to the manuscript:
>
> **"We compared alignment on four datasets against ours, which included:**
>
> **I) Comparison of 22 models on the odd-one-out similarity judgments on THINGS [1] ($\rho=0.54$).**
>
> **II) Comparison of 79 models on pairwise similarity judgments from an independent dataset [2] ($\rho=0.45$).**
>
> **III) Comparison of 79 models on two-alternative forced choice data, which is a part of the NIGHTS dataset from the DreamSim paper that was not used to train the model [3] ($\rho=0.28$).**
>
> **IV) Comparison of 25 models on alignment to the validation split of the ClickMe dataset [4,5], which was used to fine-tune the Harmonizer models ($\rho=-0.43$).**
>
> **While our metric positively correlated with the metrics from the first three datasets, there were important differences. We found a negative correlation between our metric and the ClickMe-Harmonizer feature alignment, suggesting that pixel-level alignment and semantically bound global image alignment might be at odds.**"
>
> For the odd-one-out data, we compared the models reported in the original paper [1]. For [2,3], we ran the comparisons for all the models. The ClickMe-Harmonizer alignment was only computed for supervised models, as the method requires computing gradients for ImageNet classes, which we could only do for the supervised models that had ImageNet classification heads.
>
> > the authors could take human pairwise similarity judgments and treat them as features from which to learn the task
>
> This is a great suggestion. Unfortunately, pairwise similarity judgements do not exist for THINGS. However, the ground truth features of the task were generated from an embedding optimised to predict odd-one-out similarity judgements on THINGS[7]. We report how well these models predict human choice in Figure 2. While the ground truth model ranks in the top 13, it is surpassed by several multimodal models.
>
> > I didn't see an analysis of inter-rater reliability/noise roof.
>
> The sequence of observations that were given to the participants in our tasks were randomly assigned and none were identical. Therefore it is not possible to calculate inter-rater reliability. However, the question of how good can alignment be is important.  We initially thought the models trained on the ground truth features would establish a ceiling. However, to our surprise, several models exceeded this.
>
> > why were participants with less than 50% accuracy excluded?
>
> In online studies, it is common to have noisy data, where sometimes participants don’t engage with the task faithfully. Therefore, it is common to exclude participants who perform below chance. Nevertheless, when we fit our models to all participants’s data (Rebuttal Figure 4), the $R^2$ values we obtain are highly correlated with the $R^2$ values after exclusion ($\rho=0.96$ for category learning, $\rho=0.66$ for reward). Our qualitative results stay the same, where the top-ranking models are CLIP variants. Therefore, our findings are not dependent on this filtering step.
>
>
> > why might the alignment numbers be so low?
>
> We believe there are several reasons for this. First, it is difficult to predict human choices early in the task for any model, given the noise associated with uncertainty. Indeed, when we look at $R^2$ values in the first half of the task, the maximum values are 0.05 and 0.1 for the category and the reward tasks. Whereas if we look at the second half, they go as high as 0.22 and 0.21, which is in the range that you suggested. Regardless of the split, the top-performing models are CLIP models. This is a unique difficulty of learning tasks, as unsupervised similarity judgements do not require learning and therefore a similar noise effect is not expected early in the task.
>
> Second, even when we use the ground truth features to predict human choice, we see that $R^2$ values aren’t high, which again speaks to the difficulty of the task.
>
> Lastly,  it’s difficult to compare McFadden’s $R^2$ values across tasks. Some simple learning paradigms might have best-fitting models with $R^2$ values <.2[7], whereas other similar paradigms can go up to the range you suggested [8].
>
>
> [1] Human alignment of neural network representations. Muttenthaler et al., ICLR 2023
>
> [2] Evaluating (and Improving) the Correspondence Between Deep Neural Networks and Human Representations. Peterson et al., Cognitive Science 2018
>
> [3] DreamSim: Learning New Dimensions of Human Visual Similarity using Synthetic Data. Fu et al., NeurIPS 2023
>
> [4] Harmonizing the object recognition strategies of deep neural networks with humans. Fel et al., NeurIPS 2022
>
> [5] Learning what and where to attend, Linsley et al., ICLR 2019
>
> [6] Revealing the multidimensional mental representations of natural objects underlying human similarity judgements. Hebart et al., Nat. Hum. Beh. 2020
>
> [7] Finding structure in multi-armed bandits. Schulz et al., Cognitive Psychology 2020
>
> [8] Model-Based Influences on Humans’ Choice and Striatal Prediction Errors, Daw et al., Neuron 2011

---

> > ### Comment · Reviewer_NCLd · 2024-08-13
> >
> > Thank you for the rebuttal! I appreciate the additional analyses and have increased my score accordingly.

---

> > > ### Author Response · Authors · 2024-08-14
> > >
> > > Thanks for your response and updating your score!

---

### Official Review · Reviewer_pzQG · 2024-07-12

**Soundness:** 4
**Presentation:** 3
**Contribution:** 3
**Rating:** 7
**Confidence:** 4

**Summary:**

The authors evaluate alignment between humans and 77 pre-trained neural network vision models using learning tasks. Previous work has focused on comparing alignment between humans and models with similarity judgments alone; instead, here the authors asked participants to perform simple category learning and reinforcement learning tasks. These responses were compared  with linear models fit on top of the pre-trained networks. The analysis is very thoughtful and thorough, and the article goes into detail regarding the factors that influence which models are best aligned: task accuracy, model size, training, etc.

I see this work as providing a strong contribution to understanding human vs. model alignment in vision tasks. I would like to see it published at NeurIPS, pending the concern I mentioned below.

**Strengths:**

This article has many strengths
- Kudos for comparing 77 different models
- Thoughtful and rigorous model comparison
- Detailed analysis of which factors lead models to fit the human data better
- Excellent visuals for understanding the results

**Weaknesses:**

My main concern is about the method for fitting participant responses (see question below).

The categorization task of delivering images to two dinosaurs, Julty and Folty, is a bit silly and may have had participants overthinking the task.  You may have received cleaner data with a simpler framing: learning Category A vs. Category B. But I don't see this as a major issue.

**Questions:**

I don't fully understand the methodological choices in getting the pre-trained models to predict the human responses. In the categorization task, my understanding is that a regularized logistic regression was trained to predict the gold category labels. Then, the predicted category probability was used in a logistic mixed model.

With this approach, you would seemingly lose a lot of information (potentially relevant features) in the model embeddings beyond its category prediction, before mapping this prediction to human responses. Did you consider fitting a mapping between the model features and participant responses more directly? If not, why not? I would be willing to raise my score depending on the response to this question.
EDIT: Thank you for trying this and for the explanation. I see the current approach as better justified now, and raised my score accordingly.

Also, the description of the mixed model in the appendix is not satisfying, as the variables aren't defined clearly. Please explain in the rebuttal.

Typos
"Exemplary learning curves" (pg 4)

**Limitations:**

This section is fine

---

> ### Author Rebuttal · Authors · 2024-08-06
>
> > Did you consider fitting a mapping between the model features and participant responses more directly? If not, why not?
>
> We agree with the reviewer that fitting the linear/logistic regression models directly to human choices is an interesting thought (as it has for example recently been applied in a different domain by [1]). This modelling approach typically requires a lot of data to work well. For example, [1] has used data from around 1 million human choices, whereas we have only 60 or 120 choices per participant. Hence, we expected that this approach would not work well in our setting. Nevertheless, we have now also tried to fit the embeddings directly onto human choice. However, we received much poorer fits (mean $R^2$= -0.76, max $R^2$= -0.05, min $R^2$= -3.98), as this approach does not directly capture the structure of the task and likely leads to overfitting.
>
> > Also, the description of the mixed model in the appendix is not satisfying, as the variables aren't defined clearly. Please explain in the rebuttal.
>
> Under the Behavioural Analyses section in Appendix A, we will add the following in the camera-ready version:
>
> **“We used mixed-effects logistic models for both category and reward learning analyses. For category learning, we predicted correct responses per trial, using trial number as a fixed effect and including participant-specific random effects for intercept, trial number, and assigned task rule. In the reward learning model, we predicted whether the image on the right is selected, incorporating the trial number, reward difference between images, and their interaction as fixed effects. These factors, along with the assigned task rule, were also modelled as participant-specific random effects. Both models effectively captured task structure, learning progression, and individual variability in performance.”**
>
> And under the Modelling in Appendix A, we will add the following:
>
> **“We used mixed effects logistic regression models with leave-one-out predictions to assess participant choices in both tasks. For category learning, we used logistic regression probability estimates as predictors. In the reward learning task, we used the difference in estimated rewards from linear regression models as predictors. In both cases, these predictors were included as both fixed and random effects, allowing us to account for individual differences while maintaining the group effects. These correspond to the following models in R formula notation:**
>
> **Category Learning:**
>
> ```
> human_choice ~ -1 + probability_estimate + (-1 + probability_estimate | participant)
> ```
>
> **Reward Learning**
>
> ```
> human_choice ~ -1 + estimated_reward_difference + (-1 + estimated_reward_difference | participant)
> ```
>
> **Where -1 denotes no intercept."**
>
> > The categorization task of delivering images to two dinosaurs, Julty and Folty, is a bit silly and may have had participants overthinking the task. You may have received cleaner data with a simpler framing: learning Category A vs. Category B.
>
> Thanks for raising this point. It has been shown that using “fun” cover stories improves data quality in learning tasks[2]. In our experience, this also makes the tasks more engaging, especially in online settings. Furthermore, before starting the main experiment,  participants were required to answer a few comprehension questions, and they were not allowed to participate in the main task until answering these questions correctly. Therefore, we believe participants had a good understanding of the task. We will include these questions in the Appendix for the camera-ready version.
>
> > Typos "Exemplary learning curves" (pg 4)
>
> Thanks. We will change this to **“Example learning curves”**
>
> [1] Peterson, J. C., Bourgin, D. D., Agrawal, M., Reichman, D., & Griffiths, T. L. (2021). Using large-scale experiments and machine learning to discover theories of human decision-making. Science, 372(6547), 1209-1214.
>
> [2] Feher da Silva, C., Lombardi, G., Edelson, M., & Hare, T. A. (2023). Rethinking model-based and model-free influences on mental effort and striatal prediction errors. Nat. Hum. Behav., 7(6), 956–969.

---

> ### Author Response · Authors · 2024-08-07
>
> Thank you for engaging with the review process and raising the score. We are happy we could clarify our approach and the motivation behind it.

---

### Author Rebuttal · Authors · 2024-08-06

We thank all reviewers for their constructive and helpful feedback. Their input was immensely valuable to further improve our manuscript. The reviewers’ assessment was overall positive, with each reviewer recommending an initial score of at least borderline accept:

- Reviewer pzQG saw our work “as providing a strong contribution to understanding human vs. model alignment” and mentioned that they “would like to see it published at NeurIPS” given that the pending concerns are addressed.
- Reviewer NCLd highlighted that our work is an “important and timely contribution” to the field.
- Reviewer MCn8 stated that our paper “makes substantial contributions.”
- Reviewer PLVt said that our work is a “useful asset in the human-machine comparison toolbox.”

In response to the reviewers' feedback, we have made the following major modifications to our manuscript:

- We have compared the different alignment methods on two other datasets to address the data confound of the gLocal models (Reviewers MCn8, PLVt).
- We have added an analysis comparing our proposed metric to previous alignment metrics (Reviewer NCLd).
- We have added an analysis and a discussion on fitting a mapping between the model features and participant responses (Reviewer pzQG).
- We have added additional descriptions of our model and fitting procedure to the Appendix (Reviewers pzQG, PLVt).
- We have added several new analyses, such as including participants behaving at chance level, to show that our results are robust (Reviewers NCLd, MCn8).
- We now also report average accuracies in addition to $R^2$ values (Reviewer MCn8).

We describe these and other smaller changes in detail in our responses to the individual reviews below. We again want to thank the reviewers for their time and for actively taking part in the review process.

---

> ### Author Response · Authors · 2024-08-14
> **Discussion Period Summary**
>
> Dear all,
>
> We thank all reviewers for their engagement throughout the review process. **All reviewers recommend acceptance, with an average of 6.** After our rebuttal, three reviewers [pzQG, MCn8, NCLd] increased their scores. We appreciate the reviewers' constructive input and their recognition of these improvements in their updated evaluations. Thank you all again.

---

### Decision · Program_Chairs · 2024-09-25

**Decision:**

Accept (poster)

**Comment:**

All reviewers agreed that the paper presents a strong contribution to understanding how strategies learned by today's deep neural network models of vision compare to those of humans during object classification. I recommend this paper for acceptance, with the caveat that I hope the authors can include a section in the appendix of their revision where they validate models on an established benchmark (e.g., ImageNet) to increase confidence in the implementation of each. This is particularly important with the figures created over the rebuttal period, which show mismatches in how models relate to different alignment benchmarks.